

# Genome-wide association study of yield components in spring wheat collection harvested under two water regimes in Northern Kazakhstan

Akerke Amalova[1,2]  Saule Abugalieva[1,2]  Adylkhan Babkenov[3]  Sandukash Babkenova[3]  Yerlan Turuspekov[2,4]

[1] Faculty of Biology and Biotechnology, Al-Farabi Kazakh National University, Almaty, Kazakhstan
[2] Institute of Plant Biology and Biotechnology, Almaty, Kazakhstan
[3] A.I. Barayev Research and Production Centre of Grain Farming, Shortandy, Akmola Region, Kazakhstan
[4] Faculty of Agrobiology, Kazakh National Agrarian University, Almaty, Kazakhstan

## ABSTRACT

**Background**. Bread wheat is the most important cereal in Kazakhstan, where it is grown on over 12 million hectares. One of the major constraints affecting wheat grain yield is drought due to the limited water supply. Hence, the development of drought-resistant cultivars is critical for ensuring food security in this country. Therefore, identifying quantitative trait loci (QTLs) associated with drought tolerance as an essential step in modern breeding activities, which rely on a marker-assisted selection approach.

**Methods**. A collection of 179 spring wheat accessions was tested under irrigated and rainfed conditions in Northern Kazakhstan over three years (2018, 2019, and 2020), during which data was collected on nine traits: heading date (HD), seed maturity date (SMD), plant height (PH), peduncle length (PL), number of productive spikes (NPS), spike length (SL), number of kernels per spike (NKS), thousand kernel weight (TKW), and kernels yield per $m^2$ (YM2). The collection was genotyped using a 20,000 (20K) Illumina iSelect SNP array, and 8,662 polymorphic SNP markers were selected for a genome-wide association study (GWAS) to identify QTLs for targeted agronomic traits.

**Results**. Out of the total of 237 discovered QTLs, 50 were identified as being stable QTLs for irrigated and rainfed conditions in the Akmola region, Northern Kazakhstan; the identified QTLs were associated with all the studied traits except PH. The results indicate that nine QTLs for HD and 11 QTLs for SMD are presumably novel genetic factors identified in the irrigated and rainfed conditions of Northern Kazakhstan. The identified SNP markers of the QTLs for targeted traits in rainfed conditions can be applied to develop new competitive spring wheat cultivars in arid zones using a marker-assisted selection approach.

# INTRODUCTION

Bread wheat (*Triticum aestivum* L.) is the most important cereal crop in Kazakhstan, where it is grown on over 12 million hectares. Kazakhstan produces up to 20 million tons of

Corresponding author
Yerlan Turuspekov,
yerlant@yahoo.com

common wheat annually, exporting up to 5–7 million tons (*USDA, 2018*). The country is growing of spring wheat on the territory more than 80% of the total area under wheat cultivation (http://stat.gov.kz). The average grain yield is around 1.1 tons per hectare and is constrained by abiotic stresses, including drought (*Shiferaw et al., 2013*). Drought is a prevalent stress affecting spring wheat production in Kazakhstan's northern territories, where the frequency of severe drought ranges from 15% in Kostanai to 22% in Akmola according to averaged data for 1971–2011 (*Patrick, 2017*). The impacts of climate change observed over the past 20 years has significantly increased the importance of drought as a challenge for crop management (*Skirycz & Inzé, 2010*). Still, due to the need for high grain quality, bread wheat is the major agricultural commodity in the country and the preferred choice of farmers over other crops (*Abugaliyeva & Savin, 2018*). Although irrigating crops with water can mitigate the impacts of drought, it leads to a substantial increase in growth costs. Therefore, the development of drought-resistant cultivars characterized by good productivity and high grain quality under stress will be critical for ensuring food security in the future (*Foley et al., 2011*). Drought has a significant influence on the physiological functions wheat, such as in prompting stomatal closure, decreased photosynthesis, the development of oxidative stress, and the production of toxic metabolites (*Bray, 2002*). Collectively, these changes in plant physiology lead to decreased plant height, reduced total shoot length, diminished growth rates, decreased number of tillers, reduced relative water content (*Nezhadahmadi, Prodhan & Faruq, 2013*), a decline in various grain quality parameters (*Tsenov et al., 2015*) and, ultimately, substantial yield losses (*Zhang et al., 2018*). The prerequisite for the development of new competitive drought-resistant wheat cultivars is an understanding of the genetic mechanisms associated with drought stress tolerance. Previously, various genes involved in plant drought response have been distinguished and described (*Ingram & Bartels, 1996*; *Agarwal et al., 2006*; *Wei et al., 2009*; *Huseynova, Rustamova & Mammadov, 2013*; *Nezhadahmadi, Prodhan & Faruq, 2013*; *Hassan et al., 2015*; *Kulkarni et al., 2017*). In addition, several transcription factor families associated with drought in wheat have been identified, such as ethylene response factors (ERFs), dehydration responsive element binding (DREB), and zinc finger proteins (ZFPs) (*Agarwal et al., 2006*; *Kulkarni et al., 2017*). Several known genes associated with yield components, including kernel size and weight, such as *TaTGW6, TaCwi-A1, TaSus2-2B, TaSus2-2A, TaSus1-7A, TaGW2-6A, TaGW2-6B, TaCKX6-D1, TaSAP1-A1, TaGS-D1*, and *TaGASR-A1,* were identified in wheat drought tolerance experiments using comparative genomics approaches (*Khalid et al., 2019*).

Information available on drought-responsive genes is still limited as their roles have not been thoroughly determined (*Bray, 2002*; *Nezhadahmadi, Prodhan & Faruq, 2013*). One of the critical aspects in identifying important genes associated with drought tolerance is considering the strong influence of the growth environment in which yield quantitative trait loci (QTLs) are identified with significant genotype × environment interaction (GEI). For instance, the results obtained from three different genome-wide association studies (GWAS) related to the identification of QTLs for yield performance in Europe (*Guo et al., 2017*), India (*Jaiswal et al., 2016*), and Mexico (*Sukumaran et al., 2015*), which showed different responses, and the QTLs for yield components were found to be located in
different parts of the wheat genome. The sensitivity of plants to environmental factors at crucial growth phases, which determines the tolerance to stressful factors and potential yield, can explain this outcome (*Reynolds et al., 2002*). Therefore, the success of regional projects may largely depend on local GWAS using adapted germplasms and lead to the discovery of new genetic factors that can provide plants with characteristics of drought tolerance and high yield potential in a given environment.

The characterization of germplasm is a precondition for breeding activities as it provides novel variations that can be used for the marker-assisted breeding of crops (*Nadeem et al., 2020*). The discovery of new genes for specific agronomic traits became feasible after recent breakthroughs in the development of single nucleotide polymorphism (SNP) arrays (*Cavanagh et al., 2013*; *Wang et al., 2014*; *Boeven et al., 2016*; *Sun et al., 2020*). The availability of high-throughput SNP arrays has led to the massive genotyping of wheat germplasm collections (*Allen et al., 2011*; *Wang et al., 2014*; *Sun et al., 2020*), including accessions from Kazakhstan (*Turuspekov et al., 2017*). Hence, new molecular tools have provided rich opportunities for discovering marker-trait associations (MTA) for agronomic traits via the GWAS of wheat in different parts of the world (*Rahimi et al., 2019*; *Tsai et al., 2020*), including Kazakhstan (*Turuspekov et al., 2017*; *Anuarbek et al., 2020*; *Genievskaya et al., 2020*). Some QTLs associated with various drought resistance traits in wheat have been identified using linkage mapping (*Quarrie et al., 2005*; *Verma et al., 2004*; *Tura et al., 2020*) and association mapping via GWAS (*Sukumaran et al., 2015*; *Li et al., 2019*; *Lin et al., 2019*; *Mathew et al., 2019*). The current work is the first attempt to identify drought resistance-associated QTLs under irrigated and rainfed conditions in Northern Kazakhstan using GWAS. As Northern Kazakhstan is the region where more than 80% of the wheat-growing area is concentrated in the country, the findings are important for breeding programs aimed at developing improved wheat germplasm.

## MATERIALS AND METHODS

### Phenotyping of the collection under irrigated and rainfed conditions

In this study, we considered a collection of 179 spring bread wheat accessions including 92 commercial and prospective cultivars of Kazakhstan and Russia, 86 breeding lines from Alexandr Barayev Scientific-Production Center for Grain Farming (SPCGF, Shortandy, Akmola region), and a check cultivar for the Akmola region, Tselinnaya yubileinaya (TY, Table S1). Field experiments were conducted on the experimental plots of SPCGF (51°36′09″N, 71°02′24″E, 391 m above sea level) in 2018, 2019, and 2020, both under irrigated and non-irrigated (rainfed) conditions. Each accession was grown in 1 m$^2$ complete randomized block plots composed of seven rows with 50 seeds per row in two repetitions. Field management was consistent with local practices for wheat production. Irrigation (45 mm) was applied at two critical stages: tillering and booting. The raw meteorological data registered for experiments are provided in (Raw_meteo_data.xlsx).

The nine agronomic traits associated with drought resistance and grain productivity and used for phenology and phenotyping included the heading date (HD, days), the seed maturity date (SMD, days), plant height (PH, cm), peduncle length (PL, cm), number of

productive spikes (NPS, pcs), spike length (SL, cm), number of kernels per spike (NKS, pcs), thousand kernel weight (TKW, g), kernels yield per $m^2$ (YM2, $g/m^2$).

## Genotyping of the collection

Genomic DNA was extracted from a single seedling of each individual accession using the cetyltrimethylammonium bromide (CTAB) method (*Doyle & Doyle, 1990*). The DNA concentration for each sample was adjusted to 30 ng/ μL. All samples of the 179 wheat accessions were genotyped using a 20,000 (20K) Illumina iSelect SNP array at the TraitGenetics Company (TraitGenetics GmbH, Gatersleben, Germany). A total of 8,662 polymorphic SNP markers were selected for GWAS using previously published criteria (*Miyagawa et al., 2008*). According to these criteria, markers with call rate ≥ 90%, Hardy–Weinberg equilibrium fit at P ≥ 0.001, a confidence score of 0.5, and minor allele frequency (MAF) ≥5% were considered to meet the requirements. Accessions with greater than 15% missing data were excluded from further analysis.

## Analysis of linkage disequilibrium, kinship, population structure, and statistics

The statistics for yield trials were assessed using GraphPad Prism Version 9.0 (*GraphPad Prism, 2021*). GEI was analyzed using the genotype and genotype-environment (GGE) biplot method and the Finlay and Wilkinson (FW) regression analysis in GenStat software Version 19.1 (*VSN International, 2020*). The correlation analysis was calculated using Rstudio software (*RStudio Team, 2015*)).

A model-based clustering method (admixture models with correlated allele frequencies) in STRUCTURE v.2.3.4 software (*Pritchard, Stephens & Donnelly, 2000*) was used to study the population structure of the entire collection. Five runs were conducted for each K ranging from 2 to 10 with a 100,000 burn-in length and 100,000 Markov chain Monte Carlo (MCMC) iterations. The optimal number of clusters (*K*) was chosen based on the ΔK as described by (*Evanno, Regnaut & Goudet, 2005*). The obtained values were then transformed into a population structure (*Q*) matrix.

The linkage disequilibrium (LD) in the studied collection was separately calculated for each hexaploid common wheat genome (genome A, genome B, and genome D), as well as the average LD for three genomes using Java-based TASSEL v.5.2.53 software (*Bradbury et al., 2007*). The R statistical platform was used to build a plot between the pairwise $R^2$ and the genetic distance (LD decay plot) (*RStudio Team, 2015*). TASSEL was also applied to calculate a population kinship matrix (Kin) based on the scaled identity using state (IBS) method (*Stevens et al., 2011*).

## Marker-trait association analysis

The TASSEL and mixed linear model (MLM) method with the application of *K* and *Q* matrices was used for the identification of QTLs for agronomic traits both under irrigated and rainfed conditions. The analysis was based on QTL phenotypic data for the nine traits obtained from field trials in 2018, 2019, and 2020 and their average values over three years. $P < 1 \times 10^{-3}$ was used as a significant threshold for identified MTAs. The positions and sequences of the SNP markers were obtained from the 90K Array Consensus map set of the

common wheat genome (*Wang et al., 2014*). For confirmation of the correction due to *K* and *Q* matrices, the distribution lines in each quantile-quantile (QQ) plot were analyzed. In the case of several significant MTAs positioned closely to each other, the SNP with the lowest *P*-value was chosen. MapChart v.2.32 (*Voorrips, 2002*) was used to draw a genetic map.

## Candidate gene analysis

For the search for protein-coding genes that overlapped the identified significant MTAs, the sequence for each marker was used in the BLAST tool of Ensembl Plants (*Ensembl Plants, 2020*) for comparison against the reference genome of *T. aestivum*.

# RESULTS

## Phenotypic variation and correlation analysis

The field performance of the 179 local spring wheat accessions was analyzed at the SPCGF (Northern Kazakhstan) under irrigated and rainfed conditions during three field seasons in 2018, 2019 and 2020. The two-tailed *t*- test suggested that average values in all nine studied traits were significantly different between the tested irrigated and rainfed conditions. The average PH values showed the largest difference ($P < 0.0001$) between the two tested conditions (Table 1), 73.6 cm in irrigated in contrast to 61.6 cm in rainfed conditions.

On average, YM2 declined by 5.7% under rainfed ($332.3 \pm 5.68$ g/m$^2$) compared to the irrigated ($352.3 \pm 4.30$ g/m$^2$) conditions. In total, 51 accessions exceeded the YM2 of the local standard cultivar Tselinnaya yubileinaya (TY, 374.5 g/m$^2$) under rainfed conditions, including nine accessions that outperformed the standard also under irrigated conditions (Fig. 1A). The Finlay and Wilkinson (FW) regression analysis (Fig. 1B) suggested that the YM2 of four wheat accessions, particularly WS10, WS32, WS82, and WS85, was stable in all three tested years (2018, 2019 and 2020) out of the nine accessions highlighted in the box in Fig. 1A, showing YM2 values of 400 g/m$^2$ and higher.

The analysis of the average YM2 using a scattered GGE biplot indicated that 52.8% of the total variance was explained by Principal Coordinate 1 (PC1), and 47.2% by PC2 (Fig. 2). PC1 effectively separated accessions that showed the highest yield performance in irrigated and rainfed conditions, while PC2 split the entire collection into groups with higher and lower YM2 for both conditions. The GGE biplot graph essentially confirmed the results in Fig. 1A and identified the accessions with high average YM2 under irrigated and rainfed trials as well as lines that showed high yield under both conditions (i.e., WS93 and WKZ19).

Pearson's correlation assessment in both conditions indicated that average YM2 was positively correlated with NPS and TKW (Fig. 3). Interestingly, the highest correlation value of YM2 under irrigated condition was with NPS (0.39), whereas under the rainfed condition, was with NKS (0.36). We found that earlier HD was advantageous for higher TKW under rainfed conditions, while it was not a significant factor for the yield under irrigated conditions. Under rainfed conditions, a higher PH value was not a contributing factor to YM2 (Fig. 3A). Interestingly, under rainfed conditions, HD influenced both

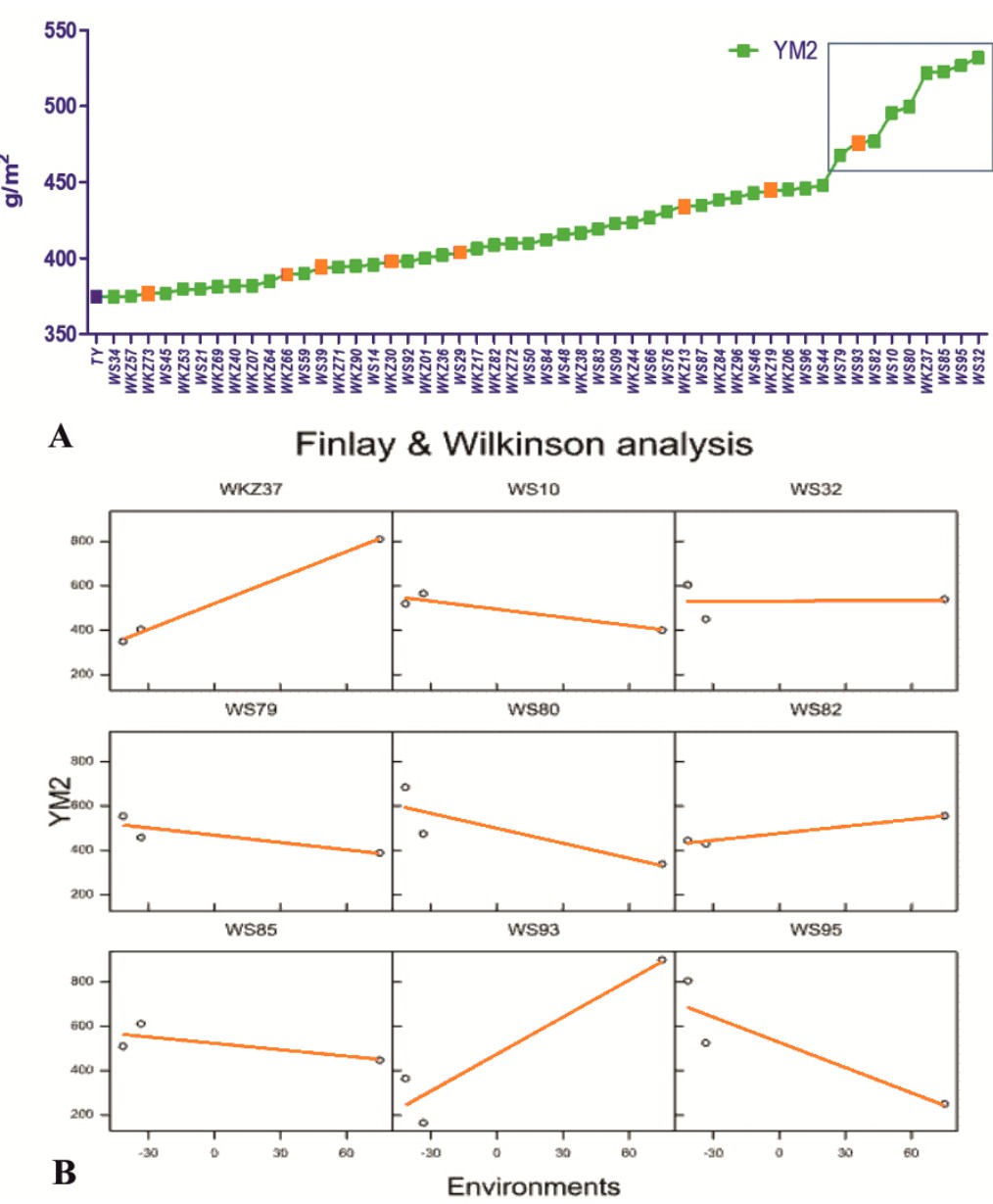

**Figure 1** **The average yield performance of the best accessions under rainfed conditions.** (A) The list of 51 spring wheat accessions that outperformed the local check cultivar Tselinnaya yubileinaya (TY) in terms of average yield per square meter (YM2) under rainfed conditions. Accessions in orange also outperformed TY in terms of average YM2 under irrigated conditions as well. The nine accessions selected in the box, with the highest average YM2, were analyzed using Finlay and Wilkinson (FW) regression. (B) The FW test suggested a different level of stability in four out of nine selected spring wheat accessions in the box in (A).

**Table 1  The significance of differences between irrigated and rainfed trials using average data in nine traits based on a two-tailed $t$-test.**

| No. | Traits | Rainfed (average) | Irrigated (average) | Significance ($P$-value) |
|---|---|---|---|---|
| 1 | Heading date (HD, days) | 48.5 ± 0.14 | 48.1 ± 0.15 | 0.00260 |
| 2 | Seed maturation date (SMD, days) | 49.3 ± 0.09 | 46.9 ± 0.10 | 1.97E−45 |
| 3 | Plant height (PH, cm ) | 61.6 ± 0.39 | 73.6 ± 0.44 | 4.4E−52 |
| 4 | Peduncle length (PL, cm) | 28.8 ± 0.25 | 32.7 ± 0.30 | 1.24E−19 |
| 5 | Spike length (SL, cm) | 9.05 ± 0.05 | 8.59 ± 0.05 | 8.18E−17 |
| 6 | Number of productive spikes (NPS, pcs) | 1.92 ± 0.03 | 2.02 ± 0.03 | 0.01322 |
| 7 | Number of kernels per spike (NKS, pcs) | 34.9 ± 0.28 | 33.6 ± 0.24 | 1.97E−06 |
| 8 | Thousand kernel weight (TKW, g) | 35.68 ± 0.19 | 37.4 ± 0.18 | 6.3E−14 |
| 9 | Yield per square meter (YM2, g/m2) | 332.3 ± 5.68 | 352.3 ± 4.30 | 0.00592 |

SL and NKS (Fig. 3). Expectedly, PL was highly correlated with PH ($P < 0.0001$), but negatively associated with NPS (Fig. 3A).

## Genetic map, population structure, and LD

The DNA genotyping data for studied 179 spring wheat accessions based on the use of the 20K SNP Illumina array resulted in the identification of 8,662 polymorphic SNP markers. The total map length for those 8,662 SNPs was 3407.6 cM, with an average chromosome length of 162.2 cM. The density of markers in chromosomes varied from 0.1 SNP per cM (chromosome 4D) to 1.7 SNP per cM (chromosome 6B).

Based on the results of the population analysis performed using STRUCTURE for the 179 accessions of wheat genotypes and STRUCTURE Harvester analyses, ΔK was optimal at K = 3. LD decay occurred at 18.5 cM (genome A), 13.1 cM (genome B), and 53.8 cM (genome D) in different genomic regions with a genome-wide LD decay of 5.0 cM at 0.1 $R^2$ (Fig. S1).

## Marker-trait associations under irrigated and rainfed conditions

The phenotypic data for nine agronomic traits of the 179 wheat accessions harvested under rainfed and irrigated conditions were subjected to GWAS using the 8,662 polymorphic SNP markers. Out of 237 total QTLs , 50 stable QTLs were identified for irrigated and rainfed conditions in the Akmola region, Northern Kazakhstan, for HD, SMD, PL, SL, NPS, NKS, and TKW, but no QTLs were detected for PH (Table 2, Table S2, Dataset S1). The highest number of stable QTLs was localized on the chromosomes of genome B (26), followed by genomes A (16) and D (8). In general, 25 QTLs were identified in both rainfed and irrigated conditions (Table 2, Fig. S2 ).

In total, eleven stable QTLs were identified for HD (Table 2). Three of them were detected only in rainfed conditions, while eight QTLs were found from the irrigated sites. One of the QTLs that was common for both tested conditions (*QHD.ta.ipbb-3B*) also affected NKS (Table 2). The largest number of QTLs identified in the rainfed trials was associated with SMD (8 of 12 total QTLs). PH and PL were one of the key traits to this study, as the two tested sites highly significantly differed in these traits ($P < 0.0001$).

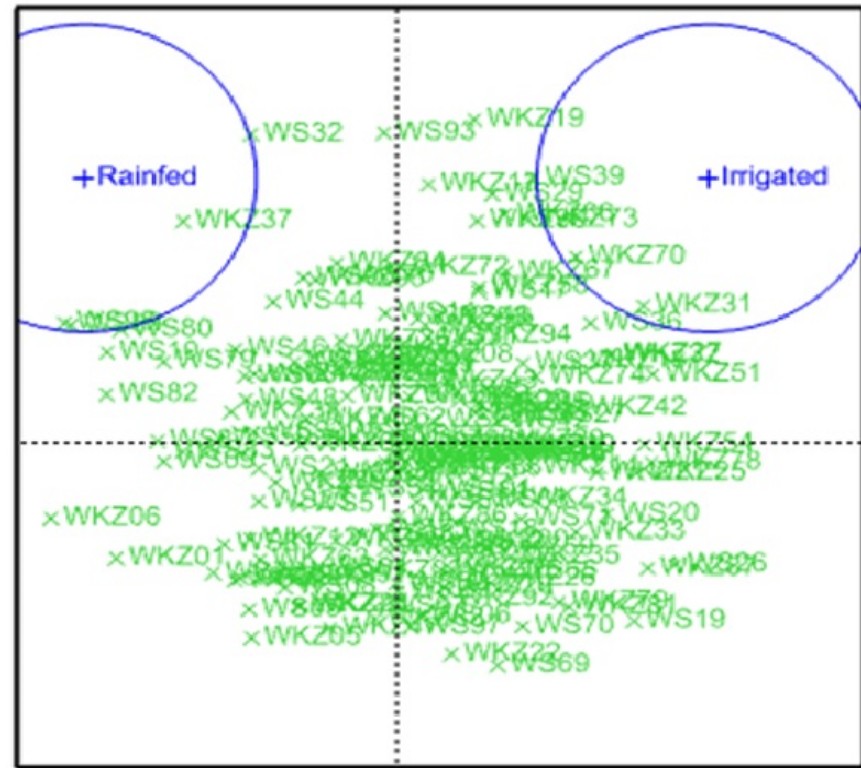

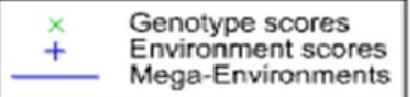

**Figure 2** **Scattered GGE biplot graph of data on averaged yield per square meter (YM2) in the collection of 179 common wheat accessions tested in irrigated and rainfed conditions (2018, 2019 and 2020).** Green and blue indicate Genotype and Environment scores, respectively.

However, only two stable QTLs (*QPL.ta.ipbb-3B.1* and *QPL.ta.ipbb-3B.2*) were identified for PL, and no stable QTL was identified for PH. The small number of QTL found under both conditions were YM2 (two QTL), NKS and SL (5 QTLs), and NPS (6 QTLs). Although all seven QTLs for TKW were identified in both tested sites, only two QTLs were detected in rainfed (*QTKW.ta.ipbb-7B*) and irrigated (*QTKW.ta.ipbb-7A*) conditions (Table 2).

Among the 46 identified QTLs, 7 pleiotropic MTAs were detected under both conditions. Those pleiotropic MTAs were mapped on the 1B, 1D, 2B, 3B, 5A, 6A, and 7A chromosomes and associated with HD (*wsnp_Ex_c8240_13914674* and *Excalibur_c20376_615*) and spike-related traits (NPS, NKS, and TKW) (Table S3, Fig. S2).

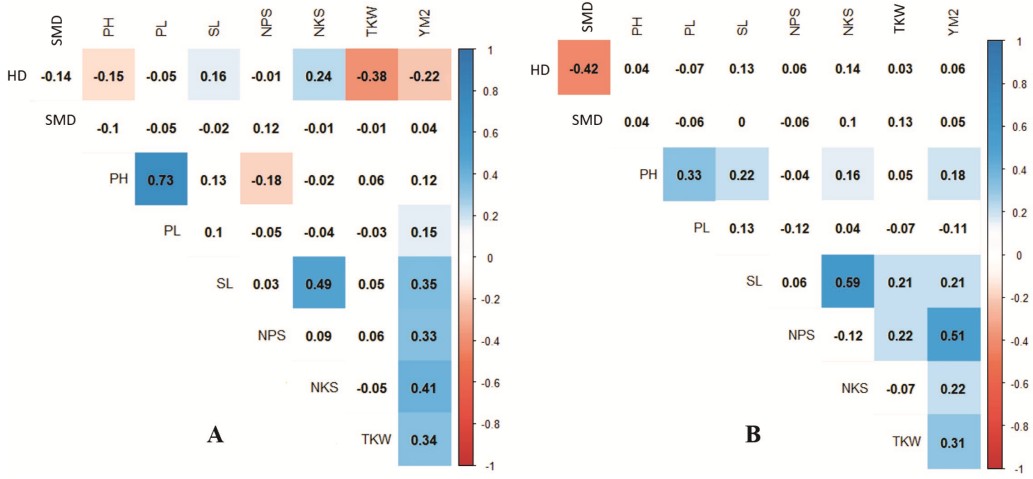

**Figure 3** Correlation analysis for the nine agronomic traits analyzed in the collection of 179 spring wheat accessions tested in rainfed (A) and irrigated (B) conditions. Blue indicates positive correlation, and red indicates negative correlation.

## DISCUSSION

### Yield assessment in the spring wheat collection under irrigated and rainfed conditions in Northern Kazakhstan

The analysis of the collection under the two tested conditions suggested substantial differences between irrigated and rainfed fields (Table 1), indicating that limitation of water supply significantly affected all nine analyzed traits in the study. Particularly, the largest difference between two sites was found for PH, which is congruent with results reported earlier (*Tsenov et al., 2015*; *Lehari et al., 2019*; *Pour-Aboughadareh et al., 2020*). The analysis of average YM2 revealed that under rainfed conditions, more than 51 common wheat accessions exceeded the YM2 of the local standard cultivar, TY, in Northern Kazakhstan (Fig. 1A). Particularly, nine accessions showed outstanding yield performance in comparison to the local check cultivar TY, and four of those lines displayed grain yield stability across all three years (Fig. 1B), suggesting that they might be an excellent source for local drought -tolerance -related breeding projects. The GGE biplot scatter method indicated that the two main principal coordinates explained 52.8% (PC1) and 47.2% (PC2) of total variability, supporting the assumption of the high diversity of the tested collection (Fig. 2). The correlation analysis for traits under rainfed conditions suggested a significantly negative correlation between HD and TKW, an important yield component, hinting that early HD might be favorable for increased yield under stressed conditions (Fig. 3A).

### Comparative QTL identification for agronomic traits in irrigated and rainfed conditions using GWAS

The GWAS analysis of the wheat collection grown under irrigated and rainfed conditions in the Akmola region of North Kazakhstan allowed the identification of 50 stable out of 237 total QTLs that were significant for eight of the studied traits (HD, SMD, PL, NPS, SL,

**Table 2** The list of QTLs for eight studied traits identified using 179 spring wheat accessions tested under irrigated and rainfed conditions of Northern Kazakhstan (2018, 2019 and 2020).

| No | QTL name | SNP | Chr | Pos | *P*-value | Effect | Irrigated | Rainfed |
|----|----------|-----|-----|-----|-----------|--------|-----------|---------|
| 1 | QHD.ta.ipbb-1B | Kukri_c39223_871 | 1B | 75.6 | 3.77E−04 | 4.84 | | + |
| 2 | QHD.ta.ipbb-2A | RAC875_c1706_1888 | 2A | 151.2 | 3.33E−04 | −1.61 | + | + |
| 3 | QHD.ta.ipbb-2B | Excalibur_c20376_615 | 2B | 76.8 | 6.54E−05 | −1.42 | + | + |
| 4 | QHD.ta.ipbb-3B | wsnp_Ex_c8240_13914674 | 3B | 32.9 | 1.49E−06 | 2.06 | + | + |
| 5 | QHD.ta.ipbb-5A.1 | BobWhite_c10385_374 | 5A | 0.00 | 1.57E−05 | −5.68 | + | + |
| 6 | QHD.ta.ipbb-5A.2 | wsnp_BF293620A_Ta_2_1 | 5A | 58.27 | 1.94E−05 | −2.13 | + | + |
| 7 | QHD.ta.ipbb-5A.3 | BS00022071_51 | 5A | 90.5 | 4.89E−05 | −1.96 | + | + |
| 8 | QHD.ta.ipbb-5B | RAC875_rep_c109634_90 | 5B | 125.0 | 3.45E−04 | −1.67 | | + |
| 9 | QHD.ta.ipbb-6A | Excalibur_c28854_1580 | 6A | 0.88 | 1.94E−06 | −6.89 | | + |
| 10 | QHD.ta.ipbb-6B | RAC875_c13610_1599 | 6B | 0.37 | 3.76E−05 | −5.39 | + | + |
| 11 | QHD.ta.ipbb-6D | Excalibur_rep_c106566_371 | 6D | 2.56 | 8.82E−06 | −6.34 | | + |
| 12 | QSMD.ta.ipbb-2A | RAC875_c57998_165 | 2A | 101.9 | 3.23E−04 | −3.66 | + | |
| 13 | QSMD.ta.ipbb-2B.1 | Kukri_c9785_1472 | 2B | 75.7 | 3.74E−04 | −0.31 | + | + |
| 14 | QSMD.ta.ipbb-2B.2 | CAP8_c5161_541 | 2B | 107.5 | 2.07E−04 | 0.47 | | + |
| 15 | QSMD.ta.ipbb-2D | Excalibur_c23239_961 | 2D | 129.0 | 1.58E−04 | 1.75 | + | |
| 16 | QSMD.ta.ipbb-3B.1 | IMX3190 | 3B | 56.6 | 5.05E−04 | −1.40 | | + |
| 17 | QSMD.ta.ipbb-3B.2 | BobWhite_c5095_634 | 3B | 69.7 | 5.05E−04 | −3.39 | | + |
| 18 | QSMD.ta.ipbb-3B.3 | BS00078844_51 | 3B | 85.0 | 5.00E−06 | −6.32 | | + |
| 19 | QSMD.ta.ipbb-3D | GENE-1805_65 | 3D | 71.9 | 6.63E−04 | −3.39 | | + |
| 20 | QSMD.ta.ipbb-4A | RAC875_c40654_206 | 4A | 120.1 | 1.76E−04 | −1.28 | | + |
| 21 | QSMD.ta.ipbb-5D | Jagger_c8037_96 | 5D | 167.0 | 6.62E−06 | −5.35 | | + |
| 22 | QSMD.ta.ipbb-6A | BS00009985_51 | 6A | 60.9 | 8.25E−05 | −5.22 | | + |
| 23 | QSMD.ta.ipbb-6B | Excalibur_c15744_322 | 6B | 0.37 | 8.66E−04 | −3.36 | + | + |
| 24 | QPL.ta.ipbb-3B.1 | wsnp_Ra_c12935_20587578 | 3B | 52.8 | 2.75E−04 | −0.26 | + | + |
| 25 | QPL.ta.ipbb-3B.2 | BS00030534_51 | 3B | 67.4 | 3.34E−04 | 4.76 | + | + |
| 26 | QSL.ta.ipbb-1A | wsnp_Ku_c1818_3557408 | 1A | 16.7 | 7.81E−04 | −0.78 | + | + |
| 27 | QSL.ta.ipbb-1B | wsnp_Ex_c26419_35667216 | 1B | 65.4 | 5.99E−04 | −0.95 | + | + |
| 28 | QSL.ta.ipbb-2B | BS00093993_51 | 2B | 108.3 | 5.64E−06 | −1.09 | + | + |
| 29 | QSL.ta.ipbb-2D | TA001453-0801 | 2D | 96.1 | 3.11E−04 | −0.62 | + | + |
| 30 | QSL.ta.ipbb-5B | Excalibur_c9391_1016 | 5B | 109.5 | 1.55E−04 | −0.78 | + | + |
| 31 | QNPS.ta.ipbb-1B | BS00078431_51 | 1B | 70.8 | 7.91E−05 | 0.31 | + | |
| 32 | QNPS.ta.ipbb-1D | BS00063511_51 | 1D | 167.1 | 8.34E−05 | 0.29 | + | |
| 33 | QNPS.ta.ipbb-2B | Excalibur_c20376_615 | 2B | 76.8 | 1.12E−05 | 0.34 | + | |
| 34 | QNPS.ta.ipbb-5A | RAC875_rep_c112818_307 | 5A | 98.9 | 3.40E−05 | −0.29 | + | |
| 35 | QNPS.ta.ipbb-6A | TA003021-1057 | 6A | 56.1 | 6.16E−04 | 0.02 | + | |
| 36 | QNPS.ta.ipbb-7A | TA003458-0086 | 7A | 133.9 | 3.54E−05 | 0.17 | + | |
| 37 | QNKS.ta.ipbb-2B | Ku_c77612_301 | 2B | 77.6 | 8.27E−05 | −4.14 | + | + |
| 38 | QNKS.ta.ipbb-3B | wsnp_Ex_c8240_13914674 | 3B | 32.9 | 1.05E−06 | 5.13 | + | |
| 39 | QNKS.ta.ipbb-4B | RAC875_c5087_310 | 4B | 71.3 | 3.28E−04 | −5.10 | + | + |
| 40 | QTKW.ta.ipbb-1B | BS00078431_51 | 1B | 70.8 | 2.75E−06 | 3.49 | + | + |
| 41 | QTKW.ta.ipbb-1D | BS00063511_51 | 1D | 167.1 | 3.64E−05 | 2.84 | + | + |

**Table 2** (*continued*)

| No | QTL name | SNP | Chr | Pos | *P*-value | Effect | Irrigated | Rainfed |
|----|----------|-----|-----|-----|-----------|--------|-----------|---------|
| 42 | QTKW.ta.ipbb-2A | wsnp_Ex_rep_c101866_87158671 | 2A | 101.9 | 4.40E−04 | 2.27 | + | + |
| 43 | QTKW.ta.ipbb-2B | Excalibur_c20376_615 | 2B | 76.8 | 1.98E−06 | 3.49 | + | + |
| 44 | QTKW.ta.ipbb-4B | Excalibur_c27349_166 | 4B | 77.9 | 2.67E−04 | −2.65 | + | + |
| 45 | QTKW.ta.ipbb-5A | RAC875_rep_c112818_307 | 5A | 98.9 | 2.23E−05 | −3.02 | + | + |
| 46 | QTKW.ta.ipbb-6A | TA003021-1057 | 6A | 56.1 | 1.67E−06 | −3.34 | + | + |
| 47 | QTKW.ta.ipbb-7A | TA003458-0086 | 7A | 133.9 | 4.56E−05 | 2.92 | + | |
| 48 | QTKW.ta.ipbb-7B | BS00063744_51 | 7B | 99.2 | 2.83E−05 | 2.68 | | + |
| 49 | QYM2.ta.ipbb-3D | BS00061125_51 | 3D | 149.8 | 3.10E−04 | 25.39 | + | |
| 50 | QYM2.ta.ipbb-7B | wsnp_Ex_c11003_17857759 | 7B | 77.2 | 5.26E−04 | 20.88 | + | |

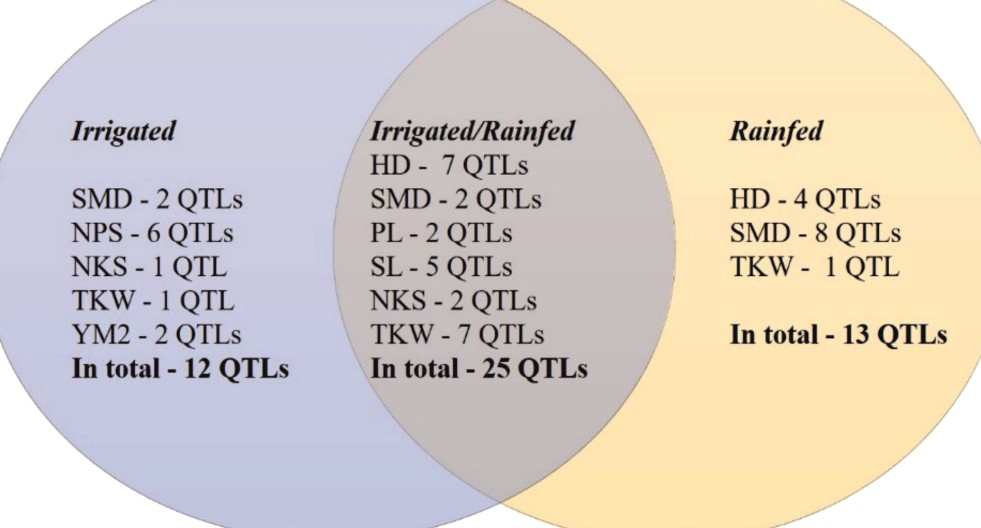

**Figure 4** Number of QTLs identified under irrigated, rainfed, and both conditions in Northern Kazakhstan in 2018, 2019 and 2020. .

NKS, TKW, and YM2) under both conditions (Table 2, Fig. S2). In total, 25 common QTLs for these traits were identified in both conditions (Fig. 4). Thirteen QTLs under rainfed conditions were revealed for HD (4), SMD (8), and TKW (1) that were not detected in trials at the irrigated conditions. Twelve QTLs were identified only under irrigated conditions for the following traits: SMD (2), NPS (6), NKS (1), TKW (1), and YM2 (2).

The assessment of QTLs for phenological traits (HD and SMD) showed that only 3 out of 23 QTLs showed effects with longer days. The remaining 20 associations were found with QTLs with shorter flowering and seed maturation time (Table 2). Two QTLs for HD, *QHD.ta.ipbb-2B* and *QHD.ta.ipbb-3B,* showed pleiotropic effects for NPS, NKS, and TKW. an SNP marker in QTL for SMD, *QSMD.ta.ipbb-2A,* was also significant in the QTL for TKW on chromosome 2A (Table 2). Interestingly, despite the stark differences between PH under irrigated and rainfed conditions, no QTL was detected in GWAS for this trait.

Evidently, the collection's accessions are fixed for this trait, and the low variation within the studied conditions was not enough to identify any MTA.

A total of eight QTLs were identified in irrigated and rainfed trials for the spike-related traits SL and NKS. Interestingly, seven of those eight QTLs were revealed both irrigated and rainfed conditions, and three QTLs were detected only under rainfed trials (Table 2). *QNKS.ta.ipbb-3B*, one of the QTLs identified only in irrigated conditions, should be particularly highlighted, as it was the factor with the highest QTL effect for the trait. Still, this QTL effect may rather be a result of the pleiotropic effect of the QTL for HD, which was identified under both conditions (Table 2). TKW is known to be one of the major yield components in wheat (*Quarrie et al., 2005*). In this study, eight of nine QTLs for TKW were detected using trials under rainfed conditions (Table 2). Particularly, *QTKW.ta.ipbb-1B* and *QTKW.ta.ipbb-2B* showed equally high QTL effects for TKW.

A comparison of the mapped QTLs analyzed in this study with those from other previous studies indicated that ten QTLs had known associations. For TKW, two QTLs *QTKW.ta.ipbb-2B.1* and *QTKW.ta.ipbb-6A* were genetically mapped close to genomic regions to QTLs for this trait identified by *Tura et al. (2020)*. Another two associations (*QNPS.ta.ipbb-1B* and *QYM2.ta.ipbb-7B*) were in the same genetic positions of QTLs identified with the analyses of six traits using GWAS based on the assessment of spring wheat in Kazakhstan (*Turuspekov et al., 2017*). *QTKW.ta.ipbb-5A* was identified in the same genetic position as in the GWAS of 285 elite spring wheat lines of wheat association mapping initiative population grown in temperate irrigated environments (*Sukumaran et al., 2015*). The only QTL identified for TKW that seems to be novel is *QTKW.ta.ipbb-7B*, which was identified under rainfed conditions and has not previously been reported. With respect to the other identified MTAs of yield components, *QNPS.ta.ipbb-2B*, *QNPS.ta.ipbb-5A*, *QSL.ta.ipbb-2D,* and *QTKW.ta.ipbb-6A* were located in close proximity to QTLs for the same traits in the study of UK reference mapping population Avalon × Cadenza in the Northern, Central, and Southern regions of Kazakhstan (*Amalova et al., 2021*).

The assessment of the identified only in rainfed conditions suggested that 12 of 13 QTLs are associated with phenological traits HD and SMD (Fig. 4), which underlines the importance of plant growth-related traits in avoiding water deficiency stress. Particularly, *QHD.ta.ipbb-6A* and *QHD.ta.ipbb-6D* for HD and seven QTLs for SMD (from chromosomes 3B to 6A) were found to be essential for early flowering and seed maturation time under rainfed conditions (Table 2). The locations of QTLs for HD compared to known flowering genes showed that the position of the most important SNP for *QHD.ta.ipbb-5A.1* completely coincides with the physical position of *Vrn1* (587,4 Mb; Table S3). *QHD.ta.ipbb-5A.3* for HD and *QSMD.ta.ipbb-3B.3* for SMD were also previously identified by *Sukumaran et al. (2015)*, confirming the robustness of the identified QTLs for HD in this study. Our literature survey indicated that the remaining nine HD and eleven SMD associations seem to be novel QTLs, as none of them were reported elsewhere (Table S3).

## Localization of significant SNPs in the identified QTLs for the studied agronomic traits in the wheat physical map

The alignment of the most significant SNPs in the 50 identified stable MTAs with sequences in the Wheat Ensembl database (https://plants.ensembl.org/Triticum_aestivum/Info/Index) suggested that SNPs in 43 and 7 MTAs were in genic and intergenic positions, respectively (Table S3). Interestingly, two SNPs in the QTLs for HD under rainfed conditions on homeological chromosomes 6A and 6B (*QHD.ta.ipbb-6A* and *QHD.ta.ipbb-6B*) were aligned with E3 ubiquitin-protein ligase (UPL) (Table S3). Similar to the above two QTL locations for HD, the homeological region on the distal part of the short arm of chromosome 6D also carries an MTA for HD (Table 2, Figs. 4A, 4B). However, the SNP in this MTA aligned to a different gene (Table S3), most probably because of poor representation of polymorphic markers in the genome D. The UPL, along with E1 ubiquitin-activating and E2 ubiquitin-conjugating enzymes, is known to participate in the ubiquitylation of proteins (*Liu et al., 2020*). Ubiquitylation is essential for the regulation of various biological processes, including growth and development, response to biotic and abiotic stress, and regulation of chromatin structure (*Ramadan et al., 2015*; *Xu et al., 2021*). Additional confirmation of the relationship between ubiquitylation and HD in this study is provided by the SNP alignment of *QHD.ta.ipbb-1B* and *QHD.ta.ipbb-3B* with the Wheat Ensembl database. Particularly, *QHD.ta.ipbb-1B* was aligned with a putative ubiquitin carrier protein, and *QHD.ta.ipbb-3B* with ubiquitin core domain-containing protein (Table S3). In other sequence alignments of the identified QTLs, the SNP in the most significant QTL for SMD (*QSMD.ta.ipbb-3B.3*) was aligned with the unknown function protein. The position of the SNP for *QTKW.ta.ipbb-1B* showed high confidence alignment with the position of a gene from *Tetratricopeptide-like helical domain superfamily*, which enables plants to cope with adverse environmental stresses and allows them to rapidly acclimate to new conditions (*Sharma & Pandey, 2016*).

Hence, the identified SNP markers for discovered 50 stable QTLs of eight agronomic traits, including eleven QTLs for HD and twelve QTLs for SMD, can be recommended for further validation tests in spring wheat projects for efficient construction of new and highly competitive cultivars in arid zones.

## CONCLUSION

The collection of spring wheat consisting of 179 local cultivars and promising lines showed a wide range of grain yield under two water regimes (irrigated and rainfed) in the Akmola region of northern Kazakhstan in 2018, 2019 and 2020. In total, 51 accessions exceeded the YM2 of the local standard cultivar Tselinnaya yubileinaya under rainfed conditions, including four accessions, WS10, WS32, WS82, and WS85, which were stable in all three tested years. The GGE biplot method was applied using two principal coordinates, and confirmed the collection's high yield variability under both tested conditions. The results of Pearson's correlation testing suggest that earlier HD is advantageous for higher TKW, which is one of the main yield components, under rainfed conditions. The SNP genotyping of the studied collection using the 20K Illumina SNP array allowed the identification

of 8,662 polymorphic SNP markers. The field phenotypic data of nine agronomic traits and polymorphic SNP data were used to identify MTA based on a GWAS. Of 237 total QTLs, 50 stable QTLs were identified in irrigated and rainfed conditions in the Akmola region, Northern Kazakhstan, by studying HD, SMD, PL, SL, NPS, NKS, TKW, and YM2. In general, 12 QTLs were identified only in irrigated, 13 QTLs only in rainfed, and 25 QTLs both rainfed and irrigated conditions. Of the 13 QTLs identified only under rainfed conditions, 12 were associated with flowering and seed maturation time, suggesting that early flowering time is essential for avoiding water deficiency stress. The literature survey indicated that nine QTLs for HD and 11 QTLs for SMD are presumably novel genetic factors identified in irrigated and rainfed conditions, and, therefore, they can be further validated for their efficiency in wheat breeding projects.

### Funding
This study was supported by (1) the grant AP08855387 "Nested association mapping for gene discovery and deployment for improvement of yield, quality, and disease resistance in bread wheat" for 2020–2022 by the Ministry of Education and Science of the Republic of Kazakhstan, and (2) BR10765056 "Creation of highly productive varieties and hybrids of grain crops based on the achievements of biotechnology, genetics, physiology, plant biochemistry for their sustainable production in various soil and climatic zones of Kazakhstan" within the framework of program-targeted financing by the Ministry of Agriculture of the Republic of Kazakhstan for 2021–2023. The funders had no role in study design, data collection and analysis, decision to publish, or preparation of the manuscript.

### Grant Disclosures
The following grant information was disclosed by the authors:
Ministry of Education and Science of the Republic of Kazakhstan for 2020–2022: AP08855387.
Ministry of Agriculture of the Republic of Kazakhstan for 2021–2023: BR10765056.

### Competing Interests
The authors declare there are no competing interests.

### Author Contributions
- Akerke Amalova performed the experiments, analyzed the data, prepared figures and/or tables, authored or reviewed drafts of the paper, and approved the final draft.
- Saule Abugalieva analyzed the data, authored or reviewed drafts of the paper, and approved the final draft.
- Adylkhan Babkenov performed the experiments, authored or reviewed drafts of the paper, and approved the final draft.
- Sandukash Babkenova conceived and designed the experiments, performed the experiments, authored or reviewed drafts of the paper, and approved the final draft.

- Yerlan Turuspekov conceived and designed the experiments, analyzed the data, authored or reviewed drafts of the paper, and approved the final draft.

## Data Availability

The raw measurements are available in the Supplementary File.

## Supplemental Information

Supplemental information for this article can be found online at http://dx.doi.org/10.7717/peerj.11857#supplemental-information.

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
