# Peer review of "Genome-wide association study of yield components in spring wheat collection harvested under two water regimes in Northern Kazakhstan"

_PeerJ, doi:10.7717/peerj.11857_

## Round 0.1 · original submission · Major Revisions

· Academic Editor

Major Revisions

After carefully evaluating the review reports, I am convinced that the article needs MAJOR REVISION. The authors need to improve English language of the paper, as suggested by the reviewers.

n

·

Basic reporting

Wheat is one of the most important crops in the world. water stress has a large effect on total grain yield and yield components. It has been widely studied in Wheat. In this work, Amalova et al. performed the genome-wide association study on the basis of 179 wheat genotypes which were genotyped based on a 20K SNP chip. They identified several QTL responsible for eight yield components and further revealed the homology of highly stable candidates. However, I do not think the current manuscript is suitable for publication. The identification of marker traits association is interesting but the writing and description quality need to improve.

Experimental design

Experimental design and analysis are well designed and described. Hence I would like to suggest adding the candidate gene annotation analysis if possible.

Validity of the findings

The results are good and innovative

Additional comments

The yield and yield components are the well-studied topics in wheat, while the review literature did not summarize the already known genes and QTLs for those yield components. hence it is sugested to add a table for known genes/QTLs for already known genetic information of yield components and cite in the introduction section.

To improve the readability of the manuscript, it needs extensive editing for linguistic corrections for spelling mistakes, ambiguous sentences, and grammatical mistakes. For example
L48 “umber”=number, L158-161 and so on.
L54 “QTLs were identified for…..” need to make clear as “identified as stable QTLs” otherwise need to write criteria that how were they identified.
L134 what A.I. stands for? It is suggested to write in full
L141: “shown in Raw data”, Please make a table number and indicate that table number here as “supplementary Table 1” etc.
The units should be in international standard format for example L181 “P<1E-03”=P < 1x10-3

Reviewer 2 ·

Basic reporting

The article is poorly written, and ambiguous words have been used throughout the manuscript.

Experimental design

The Authors used a Randomized design instead of a Randomized Black design which is recommended for the field experiments. I do not think their design is appropriate for this type of field study.

Validity of the findings

The findings are valid but the writing style does not meet the minimum requirements of scientific publications.

Additional comments

A major modification is needed to accept it for publication.

Reviewer 3 ·

Basic reporting

The present study aimed to explore the QTLs associated with yield traits in Spring wheat collection. Authors conducted experimentations in under irrigated and rainfed conditions in Northern Kazakhstan. Such types of studies are present-day requirements. I found this study informative and should be considered for publication by addressing my few minor changes.

1. Manuscript needs moderate language control.
2. In title authors are saying they performed GWAS, while in the abstract and whole manuscript, They are stating the identified QTLs. I will suggest the authors to use the word "genetic basis" in place of QTLs.
3. In the abstract Authors stated that "The study suggested
4. Please add this statement at the start of line 109: Characterization of germplasm is a prerequisite for the breeding activities as it provides novel variations that can be used for the marker-assisted breeding of crops (10.3390/genes11010036).
9 QTLs for HD and 11 QTLs for SMD were presumably novel genetic factors" I will suggest the to replace QTL with SNPs
4. I will suggest the authors construct Manhattan plots for HD, SMD, PH, PL, SL, NPS, NKS, and TKW.
5. Conclusion section should be reconsidered

Experimental design

Experimental design is fine.

Validity of the findings

Satisfactory

·

Basic reporting

The presented idea is not very clear and written in a dispersed way. The manuscript needs extensive English editing. I must suggest reviewing the manuscript from English native speaker.
The manuscript provided sufficient literature that helps the reader to know the background of the conducted study.
The manuscript structure is poor and need extensive changes. The portions (Materials and methods, Results and Discussion) of the manuscript must be presented in the order like; description of the field data, genomic DNA isolation, diversity parameters, structure analysis and finally association and MTAs analysis.
The manuscript presented the relevant results to the hypothesis.

Experimental design

The conducted study falls into the aims and scope of the journal “Peer J”.
The presented research question is well defined, relevant and meaningful. The conducted study identified genomic regions that will be beneficial to the wheat breeders working in the dry arid zones in Kazakhstan.
The proposed investigations are conducted rigorously as they repeated the field experiments for three years. Other ethical standards are not applicable to the conducted study.
The manuscript does not describe the methods with sufficient detail and are also presented randomly. Materials and method section of the manuscript need to be improved. It is suggested to perform normality test regarding the field data. It is not described that how differences among the tested accessions estimated and which software was used.

Validity of the findings

The current study reflected the identification of candidate parents for wheat breeding activities in the dry arid zones of Kazakhstan. It also identified the genomic regions related to important agronomic traits. The findings of the conducted study will pace wheat breeding activities and will be helpful in the food security.
The manuscript presented and provided all the underlying data related to support the hypothesis. The data is presented randomly in an inaccurate way. It is always suggested to present the data in a systematic way. We must present the field data first and then molecular data. In molecular data, first describe diversity parameters, STRUCTUE, UPGMA, PCoA analysis etc, and then perform QTLs/MTAs analysis. It is suggested to revise the materials and method, results and discussion section of the manuscript in the proposed pattern.
The conclusion portion of the manuscript can be improved.
The authors proposed the identified QTLs as novel factors but they are not validated yet. First they should recommend it for validation and after that they can be used in molecular wheat breeding programs.

Additional comments

Your manuscript entitled "Genome-wide association study for yield components in spring wheat collection harvested under two water regimes in Northern Kazakhstan" has now been reviewed. Overall, the manuscript does not look sound and suitable for publication in the present form and need major revision but I suggest some major changes that must be properly addressed before acceptance/publication. The manuscript provided sufficient data but presented inaccurately. The manuscript is poorly written and extensive English editing is suggested. The data in the manuscript is presented randomly in an inaccurate way. The discussion portion needs to be revised. The conducted study observed no QTL for PH although PH and 1000 seed weight are the traits that most of the times showed QTLS in most of the studies. This is very interesting to me. It is suggested to read 2 or 3 good quality papers and observe how the data is presented. Some of the minor changes are also highlighted in the manuscript that needs to be addressed properly.

---

## Round 0.2 · accepted · Accept

· Academic Editor

Accept

The authors have made significant improvements in light of the comments raised by reviewers. The article is in publishable form now and may be accepted.